# Content Analysis of Skin Cancer Screenings on Pinterest: An Exploratory Study

**DOI:** 10.3390/ijerph19052507

**Published:** 2022-02-22

**Authors:** Julie Merten, Jessica King, Ashley Dedrick

**Affiliations:** 1Department of Public Health, Brooks College of Health, University of North Florida, 1 UNF Drive, Jacksonville, FL 32224, USA; aldedrick13@gmail.com; 2Health and Kinesiology, University of Utah, 201 Presidents Circle, Salt Lake City, UT 84112, USA; jlking@wakehealth.edu

**Keywords:** skin cancer, social media, screenings, Pinterest, health communication, prevention

## Abstract

Skin cancer rates are rising in the United States, yet screening rates remain low. Meanwhile, social media has evolved to become a primary source of health information, with 40% of daily users of Pinterest reporting the platform as a “go-to” source. The objective of this research paper is to examine how skin cancer screenings were portrayed on Pinterest. Using the search terms “skin cancer screening” and “skin cancer exam”, researchers sampled every fifth pin to collect 274 relevant pins. Two researchers coded the pins, and interrater agreement was established at 94%. The results showed that twenty-two percent of the sample depicted skin cancer screening in a negative way, yet 41.5% noted that early detection leads to better outcomes. The pins were geared toward younger, white women with minimal depiction of people of color. Few pins included comprehensive information about skin cancer risk factors, importance of routine self-screenings, or what to expect with a medical provider. Fifty-eight percent of pins included links to personal blogs. In conclusion, social media has become a powerful source of health information, yet much of the posted information is incomplete. These findings present public health experts with an opportunity to disseminate more comprehensive skin cancer screening information on social media.

## 1. Introduction

Skin cancer screenings—either a self-skin examination or a total body skin examination conducted by a dermatologist—can reduce melanoma mortality rates by 65% [1,2]. Although there are no national guidelines for skin cancer screenings, the leading skin cancer prevention organizations such as the American Cancer Society and American Academy of Dermatology recommend that people conduct monthly skin examinations to look for changes [3]. Further, people at higher risk for skin cancer (family history, multiple sunburns, UVA and UVB exposure from the sun or indoor tanning, light skin and red hair) should see a dermatologist each year for a total body skin examination [2,3,4,5,6,7].

Although skin cancer screenings are quick, noninvasive, and relatively inexpensive (free for self-skin examination, and most total body skin examinations are covered by insurance), rates are low. Various studies have found rates from 9% to 18% of US adults that reported checking their skin for signs of skin cancer and roughly 20% of adults that reported having a total body skin examination by a healthcare provider [8,9,10,11,12,13,14,15,16].

Social media has evolved to become a primary source of health information, often replacing the guidance of healthcare providers. Seventy-seven percent of US adults have used search engines to search health conditions, and more than 90% of adults have reported using social media as a source of health information [17,18,19,20,21,22,23]. Social media platforms have become even more powerful sources of health information because there is a feeling of a trusted relationship with the person or company sharing the post, pin, article, or video. Among the major social media platforms (Facebook, Twitter, Pinterest, Snapchat, and Instagram), Pinterest is a highly visual bookmarking site that allows users to create their own themed boards. Users can search and share content in the form of photos, videos, recipes, or infographics, and can pin to their boards for access at any time and to share with followers. Pinterest is the third largest social media platform with more than 478 million active users and with the majority of users being female (82%) and younger than age 40 (67%); 40% of daily users reported Pinterest as a “go-to” resource for health information. Of those daily users, 84% tried something new based on information from a Pinterest post [24].

Pinterest offers tremendous potential as a source of health information, and multiple studies have examined the quality and tone of information shared on Pinterest. Specific to skin cancer, a 2019 study examined “skin tanning” specific pins and found that most pins positively portrayed tanned skin with positive reinforcement that tanned skin improved appearance [25]. Merten et al. examined 189 homemade sunscreen recipes on Pinterest and found that the homemade sunscreen recipes were positively portrayed, yet the recipes shared would not provide adequate UV exposure protection [26]. Tang and Park (2017) found that pins emphasized the role of indoor tanning and UV exposure as a primary cause of skin cancer, yet pins about non-melanoma skin cancer were lacking in information and health belief model (HBM) constructs [27].

HBM is one of the most widely used theoretical models for explaining and predicting human behavior and uses helpful constructs for examining how skin cancer screening content is portrayed on social media. The major tenets of the theory offer insight on how perceived benefits and risks of skin cancer screenings are presented and if perceived susceptibility and severity of potential skin cancer screening risks are apparent [28,29]. Further, the cues to action provide the impetus in messaging to encourage someone to act, in this case to prompt someone at risk for developing skin cancer to have a screening. Equally important in effective health behavior messaging is self-efficacy, which is someone’s belief that they can effectively get screened. McGirter and Hoffman-Goetz (2016) examined 574 articles and 905 images about skin cancer risk for HBM constructs and found minimal focus on early detection with little messaging to encourage early detection [30]. Specific to social media and Pinterest, Guidry’s work (2016) examining waterpipe smoking portrayal revealed that most of the HBM risk constructs were not portrayed, yet waterpipe smoking was generally favorably presented [31]. Another study of cigar, cigarette, and waterpipe smoking also found that college students believed that waterpipe smoking was a better choice when compared to cigars or cigarettes (perceived benefit) [32].

The representation or lack of representation of key HBM constructs offers insight for this study on how skin cancer screening risks and benefits are portrayed and if perceived susceptibility and severity of skin cancer, cues to action, and self-efficacy are present in the pins.

Presently, there are no studies specifically examining how skin cancer screening is portrayed on Pinterest.

Given the rise in use of Pinterest as a source of health information, this exploratory study seeks to answer the following research questions:How are skin cancer screenings (both self-exams and those conducted by a healthcare provider) portrayed on Pinterest?What is the quality and accuracy of the information available?What Health Belief Model constructs are apparent?How do users engage with skin cancer screening pins?

## 2. Methods

### 2.1. Sample

This study is exempt from Institutional Review Board approval because no personal or identifying information was collected, the focus of the study was on content rather than on personal demographic information. On 2 September 2020, the following keywords were entered into the Pinterest search bar: “skin cancer screening” and “skin cancer exam.” The keywords were selected after an extensive search of skin cancer screening literature and pilot searches of Pinterest using various search terms. The pilot testing process used “total body skin examination”, “skin cancer”, and “skin exam”, but those terms did not yield results related to the research question focusing on skin cancer screenings. The selected keywords yielded the most comprehensive collection of pins relevant to skin cancer screening that was pertinent to the research questions. Using methodology from the pioneer studies in social media content analysis [25,26,27,29,31,32,33], researchers used a new Pinterest account with a “fake” age of 40, the profile did not include any of the other optional profile information, including preferred pronoun, website link, or biography, and no pins were saved nor boards created. Researchers collected 1370 pins and sampled every fifth pin. Screenshots were used to capture the pin, any links, and user comments. A total of 274 pins were captured; duplicate pins (*n* = 10), pins not related to skin cancer screenings (*n* = 12), and advertisement pins such as deodorants or personal razors (*n* = 5) were excluded for a total of 247 relevant pins for analysis.

### 2.2. Coding Instrument

A codebook for skin cancer screenings was developed, tested, and used for this study using code categories from previous health-related Pinterest studies, the relevant skin cancer screening literature, including American Academy of Dermatology and American Cancer Society guidelines, and HBM constructs. Researchers created a database of all recommendations from the major skin cancer prevention organizations for self-skin cancer screenings and total body skin examinations by a health care provider. Those recommendations were converted into the codebook. See Table 1 for an abbreviated (due to space limitations) list of variables.

Methodology from the landmark studies was also used to measure engagement, including number of saves, number of followers, and number of comments [25,26,27,29,31,32,33]. HBM construct variables were operationalized and included in the codebook with dichotomous choices of yes the pins included the construct or no the pin does not include evidence of the construct.

The codebook was tested on a pilot sample of 30 pins. Two researchers coded the pins independently and then met to discuss coding discrepancies and challenges. After pilot testing modifications, the codebook was used to code the full sample.

### 2.3. Procedures

Interrater reliability was established by training two coders from the research team with one researcher coding all 247 pins and the second researcher coding 100 pins. After pilot testing modifications, coders were at 95% agreement. Disagreements in coding were resolved by discussion.

The excel file of 274 coded Pinterest pins was imported into Statistical Package for the Social Sciences v.25. (SPSS; Armonk, NY, USA, 2017) for simple descriptive analysis. The comments were collected and manually organized for themes. Thematic analysis was outside the scope of this study.

## 3. Results

### 3.1. Skin Cancer Screening Portrayal on Pinterest

The overarching research question was to examine how skin cancer screenings were portrayed on Pinterest. Pins were depicted as images with text (43.3%, *n* = 107) or images only (32.4%, *n* = 80) with 23.5% and *n* = 58 for infographics, and 0.8% and *n* = 2 for text only. The “stance” toward skin cancer screenings was mostly positive (45.3%, *n* = 112), 32.4%, *n* = 80 were balanced, yet 22.3%, *n* = negatively depicted skin cancer screenings. The appeal was predominantly health-driven (81.4%, *n* = 201) with smaller groups appealing to beauty (12.6%, *n* = 31) and fear (3.6%, *n* = 9). The pinned images were geared toward women (44.1%, *n* = 109), yet 26.7% (*n* = 66) did not have a gender appeal, 22.7%, *n* = 56 were geared toward males and females, and only 6.5%, *n* = 16 pins were toward men. Half of the pins (50.6, *n* = 125) did not have an apparent age focus, but of the pins that had images of people or mentioned ages, they were mostly geared toward young adults (30.8%, *n* = 76) and middle-aged adults (14.6%, *n* = 36) with only 4.0%, *n* = 10 toward older adults. The majority of pins did not depict any people of color (86.2%, *n* = 213). The majority of the pins did not have a clear message about the benefit of skin cancer screenings (50.8%, *n* = 125; however, 45.1% (*n* = 111) emphasized that early detection of cancer leads to better outcomes.

### 3.2. Quality and Accuracy of Skin Cancer Screening Information

The second research question sought to examine the quality and accuracy of the information in the sampled pins. Researchers used the recommendations from the American Academy of Dermatology for those who should seek skin cancer screenings and examined pins to see if those elements were included: recommendation for having an annual exam for those at increased risk for skin cancer (1.2%, *n* = 3), a specific frequency of screening (7.7%, *n* = 19), to seek care immediately for suspicious skin areas (11.4%, *n* = 28), sores than do not heal (0.0%, *n* = 0), changes in moles of freckles (13.4%, *n* = 33), inherited risk such as red hair or freckles (2.0%, *n* = 5), more than 50 moles (0.0%, *n* = 0), a family history of skin cancer (1.2%, *n* = 3), genetic sun sensitivity (2.0%, *n* = 5), sun exposure (15.9%, *n* = 39), use of indoor tanning (6.1%, *n* = 15), one blistering sunburn (0.0%, *n* = 0), precancerous conditions such as actinic keratosis (0.0%, *n* = 0), personal history of skin cancer (2.0%, *n* = 5), basal or squamous cell cancers (6.5%, *n* = 16), or prior treatments for other cancers (0.8%, *n* = 2).

Of the 247 pins, 19.9% (*n* = 49) focused on healthcare provider skin cancer screenings, and an additional 8.1% (*n* = 20) focused on both provider screenings and self-skin cancer screenings. The majority of pins (60.1, *n* = 148) did not specifically address provider or self-screenings. Of the 28%, *n* = 69 pins that showcased provider (or both) screenings, we coded the pins for information that would help a patient prepare for that examination. Variables included information indicating that the entire body would be examined (4.3%, *n* = 3), to avoid wearing nail polish and make-up (2.8%, *n* = 2), there is no machine used such as in breast cancer or colorectal screenings (0.0%, *n* = 0), a biopsy may occur during the examination (0.0%, *n* = 0), or any information about the difference in training between a primary care provider and dermatologist (0.0%, *n* = 0).

For pins focused on self-exams or pins that included both provider and self (19.9%, *n* = 59), we coded for information that should be included for a comprehensive self-examination. Variables included instructions for measuring moles (30.5%, *n* = 18), frequency of exam (10.1%, *n* = 6), directions to consult with a doctor with any unusual findings (52.5%, *n* = 31), use of good lighting (1.6%, *n* = 1), use a mirror or partner for difficult areas to reach (20.3%, *n* = 12), note the pattern of blemishes, freckles, moles, and other marks to note any changes (8.5%, *n* = 5), include the scalp (33.9%, *n* = 20) and bottoms of feet in the examination (7.3%, *n* = 18), the ABCDE of moles (45.8%, *n* = 27), other signs including new, expanding, changing growths, spots, or bumps (57.6%, *n* = 34), a sore that does not heal (0.0%, *n* = 0), a rough or scaly patch (1.6%, *n* = 1), or a wart-like growth (0.0%, *n* = 0).

### 3.3. Health Belief Model Constructs

The third research question examined which HBM constructs were apparent, and the findings are summarized in Table 2.

### 3.4. User Engagement with Skin Cancer Screening Pins

The final research question examined Pinterest user engagement with skin cancer screening pins. Of the 247 pins, 84.2%, *n* = 208 included a link to a website and of those links, 58.6%, *n* = 140 were to personal blogs, 21.3%, *n* = 51 were to a commercial website, and 19.2%, *n* = 46 were to a government health organization. The majority of pins (97.2%, *n* = 240) did not have comments. Six pins had one comment, one pin had four comments, and another pin had six comments. The comments were primarily unrelated to skin cancer screenings.

## 4. Discussion

Social media, in particular Pinterest, is a powerful source of health information. However, as with anything, the platform is only as good as the information being created and shared. This study revealed that there was substantial content about skin cancer screenings, with more than 1300 pins at the time of sampling. Of our sample of those pins, the majority of pins positively portrayed skin cancer screenings with many posts discussing the importance of early detection and encouraging users to be vigilant. Despite these uplifting findings, it was concerning to find nearly a quarter of pins with some sort of negative message about screenings with statements such as “do skin cancer screenings really work?”, “is it worth it to see a dermatologist for a total body skin examination?”, or “what your dermatologist is missing during a skin cancer exam”. These types of pins are not helpful to the overall national message to encourage people to monitor their skin for signs of skin cancer and to maintain a positive relationship with a dermatologist to monitor any changes. Messages that plant seeds of doubt are worrisome because we already have a population that demonstrates low rates of skin cancer screenings.

However, it is understandable how the general population may be confused about skin cancer screenings, given that there are not clear guidelines for routine screenings from the US Preventive Services Task Force other than to say there is not enough evidence to warrant routine screenings [33,34]. Despite the lack of national guidelines, the American Academy of Dermatology, Skin Cancer Foundation, and American Cancer Society recommend monthly self-screenings and annual provider screenings [1,4]. Some of the pins were visually appealing, with comprehensive information included in the pin and caption along with appropriate referral information to trusted sites such as the American Academy of Dermatology and Skin Cancer Foundation, along with a variety of other cancer related care centers.

The trusted skin cancer prevention and treatment organizations were the minority in our sample, with most pins having origins in personal websites related to health, holistic healing and products. This finding is particularly compelling because social media is filled with personal opinions, homemade remedies, and testimonials. The bigger question is what can we expect of non-medical professionals on social media? Do they have a responsibility to vet the information that is shared? This speaks to the current national conversation about fake news and the dangers of inaccurate or differing COVID-19 vaccine, masking, and treatment options being posted on social media. Philosophically, one would emphasize the importance of a well-educated society trained to critically analyze information more than the limiting of free speech.

It outside the scope of this study to examine if users were clicking on the links or simply reviewing the pin’s face content and continuing to scroll. The rather low percentage of comments suggest that users were not particularly interested in engaging with the pins. Social media research is rather complex in user “trust”, with more Americans reporting that they do not trust the information posted on social media. A recent survey of 1000 US consumers about locating reliable health information found that only 2% of users trust social media as a source of health information, but when that is nuanced, users trust friends and family who share information but not the colloquial term [33].

The pins were mostly geared toward young white women when examining the characteristics of the models in the pins or the shade of skin shown in the renderings of irregular moles. This also presents a reflection of a larger national public health and medical issue with appropriate skin cancer education and messaging for people of color. The existing science tells us that skin cancer is more prevalent in the white population, yet when people of color are diagnosed, it is often at a later stage with poorer outcomes [34]. There is a need for a targeted public health campaign to address people of color to encourage early detection and treatment (in addition to sun protection).

While many of the pins in this sample did not include Health Belief Model constructs, the relevant pins reflected low levels of perceived benefits of skin cancer screenings, low levels of perceived susceptibility of having skin cancer, low levels of severity of skin cancer, and minimal cues to action to prompt social media users to screen for skin cancer. The few pins that included information about perceived barriers to skin cancer screening reflected low levels of barriers to screening. Self-efficacy constructs were mostly absent from the pins.

### Limitations

This study is not without limitations. The study only provides a snapshot of the content posted and shared online through one platform. The study does not provide any information on what other sources of health information they are using to make their decisions about skin cancer screening. The study also provides no insights into if people are actually understanding or applying the health information posted on social media. A survey examining attitudes, knowledge, and behaviors of people who use social media as their primary source of health information would greatly advance this line of inquiry.

## 5. Conclusions

This exploratory content analysis of skin cancer screening pins revealed a shortage of comprehensive and accurate information posted on Pinterest. The pins were not representative of people of color, with pins primarily depicting young white women, thus reinforcing the need for social media content appropriate for people of color with images and models that are more diverse and inclusive. The sources of information were mostly personal blogs rather than validated sources of health information. Despite the critical examination of the pins, Pinterest offers an incredible platform for public health professionals and dermatologists to create comprehensive, accurate, and visually attractive social media content that links to reliable sources of information.

## Figures and Tables

**Table 1 ijerph-19-02507-t001:** Abbreviated variable codebook.

Variable	Response Options	Frequency (*n* = 247)	Scott’s Pi
Skin Cancer Screening Portrayal			
Image Type	Image only	32.4	0.97
	Image and text	43.3	
	Infographic	23.5	
Stance	Positive	45.3	0.97
	Negative	22.3	
	Balanced	32.4	
Emotional appeal	Beauty	12.6	0.93
	Health	81.4	
Gender appeal	Female	44.1	0.98
	Both	22.7	
	Not apparent	26.7	
Depicts People of Color	No *	86.2	0.99
Target age	Young adult	30.8	0.93
	Middle-aged adult	14.6	
	Not apparent	50.6	
Quality of Information			
Benefits of skin cancer examination	Benefits not apparent	50.8	0.95
	Early detection = better outcomes	45.1	
Self or provider exam	Self	10.9	0.98
	Healthcare provider	19.9	
Recommends annual exam for those at increased risk	Not included	98.8	0.98
Seek screening if you have: suspicious skin area	Not included	88.6	0.96
change in mole or freckle	Not included	86.6	0.97
Does pin mention risks: red hair and freckles	Not included	98.0	0.98
more than 50 moles	Not included	100	1.0
family history of skin cancer	Not included	98.8	0.98
frequent sun exposure	Not included	84.1	1.0
use of tanning bed	Not included	93.9	0.98
basal or squamous cancer	Not included	93.5	0.98
prior treatment/other cancers	Not included	99.2	0.98
Skin Cancer Exam with Healthcare Provider			
Informs patient entire body will be examined	Not included	95.7	1.0
Avoid nail polish and make up	Not included	97.2	1.0
Difference between primary care and dermatology	Not included	100	1.0
Self-Skin Cancer Exam			
Frequency of self-check	Not included	89.9	1.0
Discuss suspicious findings with doctor	Not included	42.5	0.98
Good lighting	Not included	98.4	0.99
Pattern of moles, blemishes, freckles, and other marks	Not included	91.5	0.98
Check scalp	Not included	66.1	0.98
Check bottoms of feet	Not included	92.7	1.0
User Engagement			
Website link	No	15.8	1.0
Comments	No	97.2	1.0
Website type (*n* = 208)	Personal blog	58.6	0.97
	Commercial site	21.3	
	Government or medical site	19.2	

* The “Yes, included on pin” option was omitted on dichotomous variables and response rates of less than 10% in this table due to space limitations.

**Table 2 ijerph-19-02507-t002:** Health Belief Model constructs.

Health Belief Model Construct	Responses
Perceived benefits of skin cancer screenings	High effectiveness (*n* = 39, 15.9%)
	Low effectiveness (*n* = 73, 29.7%)
	No information (*n* = 134, 54.5%)
Perceived barriers to skin cancer screenings	High barriers (*n* = 0. 0.0%)
	Low barriers (*n* = 40, 16.3%)
	No information (*n* = 206, 83.7%)
Perceived susceptibility of having skin cancer	High (*n* = 36, 14.6%)
	Low (*n* = 110, 44.7%)
	No information (*n* = 100, 40.7%)
Perceived severity of skin cancer	Severe (*n* = 36, 14.6%)
	Not severe (*n* = 85, 34.6%)
	No information (*n* = 125, 50.8%)
Perceived self-efficacy	Present (*n* = 44, 17.9%)
	Absent (*n* = 202, 82.1%)
Cues to action	Present (*n* = 57, 23.2%)
	Absent (*n* = 189, 76.8%)

## Data Availability

The data supporting reported results can be received upon reasonable request.

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
