# Peer review of "Content Analysis of Skin Cancer Screenings on Pinterest: An Exploratory Study"

_ijerph, 2022, doi:10.3390/ijerph19052507_

Round 1

Reviewer 1 Report

Thank you for the opportunity to review this important and timely content analysis of pins related to skin cancer screenings on Pinterest. The background for the study is a concern regarding increasing levels of skin cancer in the USA, occurring in parallel with low levels of reported rates of skin cancer screening, and a change in the use of media as a source of health information, i.e. 90% of US adults report using social media as a source of health information.

The authors examine how skin cancer screenings are portrayed on Pinterest, the quality and accuracy of the information available, how the Pins relate to Health Belief Model constructs and how users engage with the pins.

The article is in general very well-written and clearly presented. The authors also present very interesting results including how skin cancer screenings are portrayed in terms of visuals, texts and infographics; the appeal and gender and age focus of the pins; and the ethnic/racial and informational content of the pins. The authors also present important results regarding the extent to which posts on the topic connect to personal blogs, indicating that the majority of pins accessed in their study are posted by personal/individual users rather than public health related stakeholders.

At the same time, there are two overall aspects of the paper that require further development in advance of publication. These relate to the presentation of the methodology, and to the development of the discussion.

Firstly, with regards to the methodology, the authors note that the search terms were derived by reviewing relevant literature and pilot searches. More information could be provided about the literature that was reviewed and the number/results of pilot searches - i.e. how did the authors determine that their search terms were the most relevant? Furthermore, why were Health Belief Model constructs considered particularly relevant for this study?

Secondly, with regards to the methodology, the authors could provide more information about the profile of the research-based account they established to conduct the search. What was the ‘profile’ of this account with regards to age, gender, racial/ethnic profile and SES-indicators? How might this profile have influenced the content accessed and what implications might this have had for the results and discussion presented? Related to this, how did the authors determine that conducting a search through this account on one day would provide them with an appropriate sample to measure the portrayal of pins related to skin cancer screening on the Pinterest platform? And why was it considered that examining every fifth pin would be sufficient? It would seem reasonable to include the total sample of ca. 1370 pins in this primarily quantitative analysis. 

The derivation of codes in the coding book could also be more clearly explained (i.e. which codes came from where - supported by references), and to the modifications that were made. The latter is particularly interesting in the context of further research. Finally, the authors note that advertisements were excluded from their analysis. The authors could more clearly specify how they conceptualised ‘advertisements’ and what the consequences of this exclusion might be for content relating to public health campaigns and/or commercial health service providers.

Note on national context: The focus of the authors’ study is health communication in the USA. However, Pinterest is a global platform. It is unclear whether the authors considered the origin of the pins accessed and analysed in this study. Could it be that pins conflict with the ‘overall national message’ referred to by the authors because they are posted by users from different countries? This national context could also be more clearly communicated in the presentation of statistics regarding the Pinterest user-base. Do these relate to the global user base or the national user base?

Discussion:

The discussion as currently presented does not significantly synthesise or consider the implications of the results, or how these relate to previous research. The study is justified by a concern about the lack of screenings in the USA, coupled with the fact that 90% of adults report using social media to access health information. The results reveal further concerns relating to the quality of information that is available in Pinterest.

This discussion could further emphasise how the portrayal of health information in these pins relate to their sources (i.e. who posted them and whether these were private/individual accounts of public health policy stakeholders), and how this in turn relates to the quality of information (or trust in information). Furthermore, the consequences and potential implications of user engagement with the pins is not sufficiently discussed.

The authors also present important results regarding ‘people of color’. However it is not clear how the results presented relate to the sources of the pins - i.e. the user-base/posters, or how the results might relate to the way in which the research-based account was set-up.

Regarding the extent to which Pinterest is an important potential platform for the dissemination of health information, the authors could more substantially discuss how their own results present some tensions related to this, i.e. the majority of Pins they accessed were posted by personal/individual users. To what extent can individual users be expected to provide comprehensive information regarding the importance of skin cancer screening? This also relates to and has consequences for how the Pinterest platform is understood and used. 

Connected to this, and regarding the results with respect to the absence of comprehensive information in pins accessed, it would appear that the genres of communication facilitated by Pinterest present restrictions for the kind of information that can be posted. The authors note that the platform is ‘a highly visual bookmarking site’ and find that the majority of pins consist of images with text or images only, with only a quarter of images presented as infographics which might allow for a more comprehensive presentation of information.

Regarding the claim that social media ‘have become even more powerful sources of health information because there is a feeling of a trusted relationship with the person or company sharing the post…’. This should be supported by a reference to relevant literature, and perhaps discussed further given the apparent importance of trust in health communication on social media which provides the justification and context for this study.

Finally, ethical considerations are not mentioned in the article. In spite of this many of the pins posted link to individual blogs. The authors could provide information abut the extent to which their study involved accessing or storing personal (and potentially sensitive) health information, and whether they considered it necessary to inform users of Pinterest about their study, and/or to seek informed consent. The Association of Internet Researchers Ethical Guidelines is a useful resource to consult in this regard.

Minor issues:

Pg. 2, line 52-53 - statement re the potential of Pinterest as a source of communication of health information - this could be more clearly connected to previous studies referred to on the topic

Pg. 2, line 60 - what does the abbreviation NMSC stand for?

Author Response

Thank you for your valuable, thoughtful and kind feedback. Our revisions are below.

Thank you for the opportunity to review this important and timely content analysis of pins related to skin cancer screenings on Pinterest. The background for the study is a concern regarding increasing levels of skin cancer in the USA, occurring in parallel with low levels of reported rates of skin cancer screening, and a change in the use of media as a source of health information, i.e. 90% of US adults report using social media as a source of health information.

The authors examine how skin cancer screenings are portrayed on Pinterest, the quality and accuracy of the information available, how the Pins relate to Health Belief Model constructs and how users engage with the pins.

The article is in general very well-written and clearly presented. The authors also present very interesting results including how skin cancer screenings are portrayed in terms of visuals, texts and infographics; the appeal and gender and age focus of the pins; and the ethnic/racial and informational content of the pins. The authors also present important results regarding the extent to which posts on the topic connect to personal blogs, indicating that the majority of pins accessed in their study are posted by personal/individual users rather than public health related stakeholders.

At the same time, there are two overall aspects of the paper that require further development in advance of publication. These relate to the presentation of the methodology, and to the development of the discussion.

Firstly, with regards to the methodology, the authors note that the search terms were derived by reviewing relevant literature and pilot searches. More information could be provided about the literature that was reviewed and the number/results of pilot searches - i.e. how did the authors determine that their search terms were the most relevant? Furthermore, why were Health Belief Model constructs considered particularly relevant for this study?

-We added the following content:

HBM is one of the most widely-used theoretical models for explaining and predicting human behavior and has helpful constructs for examining skin cancer screening content is portrayed on social media. The major tenets of the theory offer insight on how perceived benefits and risks of skin cancer screenings are presented and if perceived susceptibility and severity of potential skin cancer screening risks are apparent [28]. Guidry’s work (2016) examining waterpipe smoking portrayal on Pinterest revealed most of the HBM risk constructs were not portrayed yet waterpipe smoking was generally favorably presented [29]. Another study of cigar, cigarette, and waterpipe smoking also found that college students believed waterpipe smoking was a better choice when compared to cigars or cigarettes (perceived benefit) [30}.

The representation or lack of representation of key HBM constructs offers insight for this study on how skin cancer screening risks and benefits are portrayed. Further, if perceived susceptibility and severity of skin cancer are present in the pinsThe pilot testing process used “total body skin examination” “skin cancer” and “skin exam” but those terms did not yield salient results.

Secondly, with regards to the methodology, the authors could provide more information about the profile of the research-based account they established to conduct the search. What was the ‘profile’ of this account with regards to age, gender, racial/ethnic profile and SES-indicators? How might this profile have influenced the content accessed and what implications might this have had for the results and discussion presented? Related to this, how did the authors determine that conducting a search through this account on one day would provide them with an appropriate sample to measure the portrayal of pins related to skin cancer screening on the Pinterest platform? And why was it considered that examining every fifth pin would be sufficient? It would seem reasonable to include the total sample of ca. 1370 pins in this primarily quantitative analysis. 

-We clarified some of the Methodology language:

Using methodology from the pioneer studies in social media content analysis [32,33,34,35,36], researchers used a new Pinterest account with the “fake” age of 40 to collect 1,370 pins and sampled every fifth pin.

We didn’t include the following language in the paper but wanted to explain our some of our sample methodology decisions. Most of the seminal articles and our various social media articles pull a sample rather the entire sample. And as far as the one day sample, in our preliminary examination of the hashtags, we didn’t see a tremendous amount of variation and new pins over the course of our two month investigative period. Happy to include this in the paper if you see fit – obviously written more professionally and less conversational.

The derivation of codes in the coding book could also be more clearly explained (i.e. which codes came from where - supported by references), and to the modifications that were made. The latter is particularly interesting in the context of further research. Finally, the authors note that advertisements were excluded from their analysis. The authors could more clearly specify how they conceptualised ‘advertisements’ and what the consequences of this exclusion might be for content relating to public health campaigns and/or commercial health service providers.

-We added clarifying language to more clearly explain the creation of the codebook from the prominent skin cancer prevention and treatment leaders: Researchers created a database of all recommendations from the major skin cancer prevention organizations for self-skin cancer screenings and total body skin examinations by a health care provider. Those recommendations were converted into the codebook

We added the following line to clarify the unrelated advertisements: A total of 274 pins were captured and duplicate pins (n=10), pins not related to skin cancer screenings (n= 12), or advertisement pins such as deodorants or personal razors (n=5) were excluded for a total of 247 relevant pins for analysis.   

Note on national context: The focus of the authors’ study is health communication in the USA. However, Pinterest is a global platform. It is unclear whether the authors considered the origin of the pins accessed and analysed in this study. Could it be that pins conflict with the ‘overall national message’ referred to by the authors because they are posted by users from different countries? This national context could also be more clearly communicated in the presentation of statistics regarding the Pinterest user-base. Do these relate to the global user base or the national user base?

-Our research team has historically kept our Pinterest studies focused on the US audience because the US has around 90 million daily users whereas the next most active country is Brazil with 30 million. We can certainly add this information if the reviewer thinks it would be helpful.

Discussion:

The discussion as currently presented does not significantly synthesise or consider the implications of the results, or how these relate to previous research. The study is justified by a concern about the lack of screenings in the USA, coupled with the fact that 90% of adults report using social media to access health information. The results reveal further concerns relating to the quality of information that is available in Pinterest.

This discussion could further emphasise how the portrayal of health information in these pins relate to their sources (i.e. who posted them and whether these were private/individual accounts of public health policy stakeholders), and how this in turn relates to the quality of information (or trust in information). Furthermore, the consequences and potential implications of user engagement with the pins is not sufficiently discussed.

-It is worth noting that some of the pins were visually appealing with comprehensive information included in the pin and caption along with appropriate referral information to trusted sites like the American Academy of Dermatology and Skin Cancer Foundation along with a variety of other cancer related care centers.  

The trusted skin cancer prevention and treatment organizations were the unfortunate minority in our sample with most pins having origins in personal blogs relating to health, holistic healing and products. It is unknown if users were clicking on the links or simply reviewing the pin’s face content and continuing to scroll. The rather low percentage of comments suggest that users were not particularly interested in engaging with the pins.

The authors also present important results regarding ‘people of color’. However it is not clear how the results presented relate to the sources of the pins - i.e. the user-base/posters, or how the results might relate to the way in which the research-based account was set-up.

-Pinterest doesn’t require disclosure of race and our “fake” collection account had no other searches or indicators of interest.

Regarding the extent to which Pinterest is an important potential platform for the dissemination of health information, the authors could more substantially discuss how their own results present some tensions related to this, i.e. the majority of Pins they accessed were posted by personal/individual users. To what extent can individual users be expected to provide comprehensive information regarding the importance of skin cancer screening? This also relates to and has consequences for how the Pinterest platform is understood and used. 

-This point greatly strengthened our Discussion and we are grateful to the reviewer for this suggestion:

The trusted skin cancer prevention and treatment organizations were the unfortunate minority in our sample with most pins having origins in personal websites relating to health, holistic healing and products. This finding is particularly compelling because social media is filled with personal opinions, homemade remedies, and testimonials. The bigger question is what can we expect of non-medical professional on social media? Do they have a responsibility to vet the information that is shared? This speaks to the current national conversation about fake news and the dangers of inaccurate – or differing – Covid-19 vaccine, masking, and treatment options being posted on social media. Philosophically, one would emphasize the importance of a well-educated society trained to critically analyze information more than limiting free speech.

It outside the scope of this study to examine if users were clicking on the links or simply reviewing the pin’s face content and continuing to scroll. The rather low per-centage of comments suggest that users were not particularly interested in engaging with the pins. Social media research is rather complex in user “trust” with more Americans reporting they do not trust the information posted on social media. A recent survey of 1,000 US consumers about locating reliable health information found that only 2% of users trust social media as a source of health information but when that is nuanced, users trust friends and family who share information but not the colloquial term [31].

Connected to this, and regarding the results with respect to the absence of comprehensive information in pins accessed, it would appear that the genres of communication facilitated by Pinterest present restrictions for the kind of information that can be posted. The authors note that the platform is ‘a highly visual bookmarking site’ and find that the majority of pins consist of images with text or images only, with only a quarter of images presented as infographics which might allow for a more comprehensive presentation of information.

Regarding the claim that social media ‘have become even more powerful sources of health information because there is a feeling of a trusted relationship with the person or company sharing the post…’. This should be supported by a reference to relevant literature, and perhaps discussed further given the apparent importance of trust in health communication on social media which provides the justification and context for this study.

-We’ve added the following content: It outside the scope of this study to examine if users were clicking on the links or simply re-viewing the pin’s face content and continuing to scroll. The rather low percentage of comments suggest that users were not particularly interested in engaging with the pins. Social media research is rather complex in user “trust” with more Americans reporting they do not trust the information posted on social media. A recent survey of 1,000 US consumers about locating reliable health in-formation found that only 2% of users trust social media as a source of health information but when that is nuanced, users trust friends and family who share information but not the colloquial term [40].

Finally, ethical considerations are not mentioned in the article. In spite of this many of the pins posted link to individual blogs. The authors could provide information abut the extent to which their study involved accessing or storing personal (and potentially sensitive) health information, and whether they considered it necessary to inform users of Pinterest about their study, and/or to seek informed consent. The Association of Internet Researchers Ethical Guidelines is a useful resource to consult in this regard.

-We added this information to start the Methods section: This study is exempt from Institutional Review Board approval because no personal or revealing information was collected, the focus of the study was content rather than personal demographic information.

Minor issues:

Pg. 2, line 60 - what does the abbreviation NMSC stand for?

-Non-Melanoma Skin Cancer – the abbreviation has been removed.

Reviewer 2 Report

The article entitled Content Analysis of Skin Cancer Screenings on Pinterest is a fairly well-planned, simple, observational study.
The assumptions and background of the problem, goals and methodology were described correctly.
The results of the study were presented clearly, correct conclusions were drawn.
In my opinion, however, the article does not fall within the thematic scope of the IJERPH journal. I suggest the authors try to publish the results of their research in sociological journals.
I do not find the subject of the article compatible with the issues of environmental health or properly understood public health.

Author Response

Thank you for taking the time to review our article. We appreciate your feedback.

Reviewer 3 Report

Page 1, last paragraph (lines 39-41), where you say “75% of US adults have used search engines… and more than 90% have used social media for health information,” please cite a source or two.

Page 2, first paragraph (lines 46-51), where you report statistics about Pinterest users, please cite a source. Ideally, please cite a recent one.

Page 2, last paragraph (lines 89-93), where you mention previous studies, are they [32 to 36] as indicated on page 3, line 98? If so, you might want to report them in line 90 on page 2.

Your introduction also serves as a brief literature review. I don’t think it is sufficient. Also, the results show only raw numbers and percentages. Without a solid theoretical foundation and statistical analysis (such as Chi-square), this manuscript reads like a data-driven news report instead of a genuine research study.

Here are a few recommendations for your consideration. First, in your literature review, you might want to begin with a media effects theory or two to explain why it is important to analyze health information on social media. Second, review studies about health information on social media to establish the link between social media use, learning health information, and the influence of such information on behavior. Third, review a few content analysis studies [such as 32 to 36] related to skin cancer and/or Pinterest, then discuss their coding categories, especially the ones you adopt. Fourth, review the Health Belief Model, including its origin, advantages, landmark studies, and why it is relevant to your study.

Regarding your results related to RQ2, you might want to develop a system to compare accurate and inaccurate information. I think a simple chi-square should work in this case and (elsewhere).

To make this a more sophisticated quantitative study, in addition to running chi-squares, please think in terms of independent and dependent variables when you develop your research questions (based on a solid literature review). This approach might lead to more advanced statistical tests than chi-squares.

Author Response

Thank you for taking the time to review our article. We are appreciative of your feedback, truly. Revisions noted below:

Page 1, last paragraph (lines 39-41), where you say “75% of US adults have used search engines… and more than 90% have used social media for health information,” please cite a source or two.

-Thank you for catching that. We have added two sources.

Page 2, first paragraph (lines 46-51), where you report statistics about Pinterest users, please cite a source. Ideally, please cite a recent one.

-When correcting this, we realized that our sources were somehow lost in the text formatting  – throughout the entire paper. We’ve added the original citations back and corrected

Page 2, last paragraph (lines 89-93), where you mention previous studies, are they [32 to 36] as indicated on page 3, line 98? If so, you might want to report them in line 90 on page 2.

-Same issue above, there was some terrible fiasco with the reference management system. We have manually checked each reference.

- we’ve added the following content to the Introduction within reason for the length of the article:

Pinterest offers tremendous potential as a source of health information and multiple studies have examined the quality and tone of information shared on Pinterest. Specific to skin cancer, a 2019 study examined “skin tanning” specific pins and found most pins positively portrayed tanned skin with positive reinforcement that tanned skin improved appearance [25]. Merten et al examined 189 homemade sunscreen recipes on Pinterest and found the homemade sunscreen recipes were positively portrayed yet the recipes shared would not provide adequate UV exposure protection [26]. Tang and Park (2017) found that pins emphasized the role of indoor tanning and UV exposure as a primary cause of skin cancer yet pins about non-Melanoma Skin Cancer were lacking in information and Health Belief Model (HBM) constructs [27].

HBM is one of the most widely-used theoretical models for explaining and predicting human behavior and has helpful constructs for examining skin cancer screening content is portrayed on social media. The major tenets of the theory offer insight on how perceived benefits and risks of skin cancer screenings are presented and if perceived susceptibility and severity of potential skin cancer screening risks are apparent [28]. Guidry’s work (2016) examining waterpipe smoking portrayal on Pinterest revealed most of the HBM risk constructs were not portrayed yet waterpipe smoking was generally favorably presented [29]. Another study of cigar, cigarette, and waterpipe smoking also found that college students believed waterpipe smoking was a better choice when compared to cigars or cigarettes (perceived benefit) [30}.

The representation or lack of representation of key HBM constructs offers insight for this study on how skin cancer screening risks and benefits are portrayed. Further, if perceived susceptibility and severity of skin cancer are present in the pins.

Round 2

Reviewer 1 Report

Thank you for the opportunity to review this revised manuscript! 

The authors have done significant work to incorporate the comments and feedback provided.

I would like to see some additional changes to the manuscript in advance of publication. These relate most importantly to the methodology.

It is still difficult to understand why the authors chose the search terms that they did. They present additional search terms used in the pilot phase and state that the results of these terms were 'not salient' - but how was this salience determined?

More information could also be provided on why the sample was considered sufficient - it is clear that the authors refer to other studies that have used similar methods - but why, in this case, was the sample sufficient?

Regarding the 'fake' account - did using a fake account present any ethical considerations for the authors? Also, Pinterest as a platform will tailor content to the profiles of user accounts- so it would be important to reflect on this in the context of the methodology - i.e. what consequences does profiling/algorithmic curation of content have for the content sampled and for the subsequent analysis.

An additional sentence or two in each case would be sufficient.

Regarding the national context and the size of the user-base in the USA - the concern I would have is that the content accessed by the fake account could have originated outside of the USA and that this would have consequences for the author's focus on the national/US concern about skin cancer and the requirement to promote screening. I.e. pins created outside of the US come from different national contexts. This has consequences for the way in which they are constructed - and therefore potentially implications for the analysis.

The discussion is improved. I would suggest the authors remove the word 'unfortunately' from the revised text - or at least explain why they consider it 'unfortunate' that very little pins were found from public health stakeholders/trusted sources of information. (This in context of the nature of Pinterest as a platform).

Regarding the collection of personal data and ethical considerations - I am not sure about the basis on which the exemption was granted as I work primarily in a European context - but it is likely that Pins posted by private individuals would also constitute personal data. Furthermore, it would be interesting to know whether the authors took steps to ensure that their data did not contain information that was private or personally revealing, or whether this is something that is more characteristic of the content posted on the platform.

Thank you again for the opportunity to review this important study.

Author Response

Thank you for your attention to our paper - it is VERY appreciated. The paper is much stronger now. Our changes per your recommendation are below.

It is still difficult to understand why the authors chose the search terms that they did. They present additional search terms used in the pilot phase and state that the results of these terms were 'not salient' - but how was this salience determined?

Response and revision:

We spent considerable time pulling language from all of the major skin cancer prevention professional organizations and skin cancer screenings was the most commonly used term. Clinicians use the term “total body skin examination” but that term is not a commonly used and when searching in Pinterest, yielded ZERO relevant pins. Relevant being defined as related to skin cancer.

We added this text to the Methods The pilot testing process used “total body skin examination” “skin cancer” and “skin exam” but those terms did not yield results related to the research question focused on skin cancer screenings.

More information could also be provided on why the sample was considered sufficient - it is clear that the authors refer to other studies that have used similar methods - but why, in this case, was the sample sufficient?

Response and revision:

With a content analysis of this depth, it would not be possible for us to closely examine and code 1300 pins. We did not feel we would yield any additional depth from analyzing all pins.

Regarding the 'fake' account - did using a fake account present any ethical considerations for the authors? Also, Pinterest as a platform will tailor content to the profiles of user accounts- so it would be important to reflect on this in the context of the methodology - i.e. what consequences does profiling/algorithmic curation of content have for the content sampled and for the subsequent analysis.

An additional sentence or two in each case would be sufficient.

Response and revision:

We did not perceive any ethical considerations of a fake account when we considered the importance of creating a new account to avoid profiling of our personal accounts.

We added the following information for clarity

Using methodology from the pioneer studies in social media content analysis [25, 26, 27, 28,29, 30], researchers used a new Pinterest account with the “fake” age of 40, the profile did not include any of the other optional profile information including preferred pronoun, website link, or biography and no pins were saved, and no boards were created.

Regarding the national context and the size of the user-base in the USA - the concern I would have is that the content accessed by the fake account could have originated outside of the USA and that this would have consequences for the author's focus on the national/US concern about skin cancer and the requirement to promote screening. I.e. pins created outside of the US come from different national contexts. This has consequences for the way in which they are constructed - and therefore potentially implications for the analysis.

Response and revision: This question piqued my curiosity as well. I examined the skin cancer screening recommendations from a handful of other countries including Australia, New Zealand, Germany, the Netherlands and the UK and found that the major tenets of skin cancer screenings (both self-screenings and provider) are quite similar and don’t contradict the major research questions of this article.

The discussion is improved. I would suggest the authors remove the word 'unfortunately' from the revised text - or at least explain why they consider it 'unfortunate' that very little pins were found from public health stakeholders/trusted sources of information. (This in context of the nature of Pinterest as a platform).

Response and revision:

Unfortunate was removed.

Regarding the collection of personal data and ethical considerations - I am not sure about the basis on which the exemption was granted as I work primarily in a European context - but it is likely that Pins posted by private individuals would also constitute personal data. Furthermore, it would be interesting to know whether the authors took steps to ensure that their data did not contain information that was private or personally revealing, or whether this is something that is more characteristic of the content posted on the platform.

Response and revision:

We operate from this IRB guidance for this study and our previously published studies using the same methodology:

Research involving collection or study of existing data, documents, and records can be exempted under Category 4 of the federal regulations if: (i) the sources of such data are publicly available; or (ii) the information is recorded by the investigator in such a manner that subjects cannot be identified, directly or through identifiers linked to the subjects.

When we collect the pins, the content is extracted without the Pinterest users handle (identity). The pin content (image) did not include personal identifiers like other platforms like Instagram which include embedded tags.

Reviewer 2 Report

Still, the article does not fall within the scope of the subject of the IJERPH journal. I propose that the authors try to publish the results of their research in sociological journals.
In my opinion, the subject of the article does not concern the issues of environmental health or properly understood public health.
The authors did not refer to my previous review. They did not try to convince me of the article's compliance with the subject of the journal.

Author Response

In the original review, we did not respond because we felt that was a judgement for the Editor to make - the suitability of a paper topic for a journal.

When selecting a journal for our work, we reviewed the Aims and Scopes of several journals. Our article is grounded in public health theory but pulls in environmental components given the role of the environment in skin cancer. Per the Aims of the journal noted below, we thought the journal was an excellent fit.

Therefore, IJERPH focuses on the publication of scientific and technical information on the impacts of natural phenomena and anthropogenic factors on the quality of our environment, the interrelationships between environmental health and the quality of life, as well as the socio-cultural, political, economic, and legal considerations related to environmental stewardship, environmental medicine, and public health.

Reviewer 3 Report

The authors revised part of the literature review, which is appreciated. However, reviewing three studies on Pinterest is not sufficient. The discussion of the Health Belief Model is not deep enough. As a result, the research questions are not well justified. To make this study publishable, the authors are urged to lengthen the literature review in order to strengthen the theoretical foundation. 

Author Response

We appreciate the feedback and love the opportunity to expand on HBM. We apologize for the brevity, in other papers, we were asked to significantly reduce our discussion of HBM in the Introduction. We've expanded the content to include the following:

HBM is one of the most widely-used theoretical models for explaining and predicting human behavior and has helpful constructs for examining how skin cancer screening content is portrayed on social media. The major tenets of the theory offer insight on how perceived benefits and risks of skin cancer screenings are presented and if perceived susceptibility and severity of potential skin cancer screening risks are apparent [28, 29]. Further, the cues to action provide the impetus in messaging to encourage someone to act, in this case to prompt someone at risk for developing skin cancer to have a screening. Equally important in effective health behavior messaging is self-efficacy which is someone’s belief they can effectively get screened. McGirter and Hoffman-Goetz (2016) examined 574 articles and 905 images about skin cancer risk for HBM constructs and found minimal focus on early detection with little messaging to encourage early detection [30]. Specific to social media and Pinterest, Guidry’s work (2016) examining waterpipe smoking portrayal revealed most of the HBM risk constructs were not portrayed yet waterpipe smoking was generally favorably presented [31]. Another study of cigar, cigarette, and waterpipe smoking also found that college students believed waterpipe smoking was a better choice when compared to cigars or cigarettes (perceived benefit) [32].

The representation or lack of representation of key HBM constructs offers insight for this study on how skin cancer screening risks and benefits are portrayed. Further, if perceived susceptibility and severity of skin cancer, cues to action, and self-efficacy are present in the pins.